# Tissue Oxygen Saturation Change on Upper Extremities After Ultrasound-Guided Infraclavicular Brachial Plexus Blockade; Prospective Observational Study

**DOI:** 10.3390/medicina55060274

**Published:** 2019-06-14

**Authors:** Mahmut Alp Karahan, Orhan Binici, Evren Büyükfırat

**Affiliations:** Department of Anesthesiology and Reanimation, Harran University Medical Faculty, 63000 Sanliurfa, Turkey; orhan_binici@windowslive.com (O.B.); evrenbf@gmail.com (E.B.)

**Keywords:** tissue oxygen saturation, near-infrared spectroscopy, ultrasound, infraclavicular block, regional anesthesia

## Abstract

*Background and Objective:* The aim of this study was to investigate whether tissue oxygen saturation (StO2) is a reliable and objective method for assessing the adequacy of infraclavicular block and to describe the time course of StO2 changes. *Materials and Methods:* In this prospective observational study, StO2 was measured in 40 patients planned for elective hand surgery under infraclavicular block. Noninvasive StO2 monitoring was used prior to ultrasound-guided infraclavicular brachial plexus block and during the first 30 min of the blockade. Sensory and motor blocks were evaluated every 5 min followed by pinprick testing and Bromage scale. *Results:* Preanesthetic median StO2 values of the blocked side and nonblocked side were similar (*p* = 0.532), whereas the postanesthetic values of the blocked side were higher. At the fifth minute and the following minute, measurements compared to the nonblocked side (*p* < 0.001). The median StO2 values increased significantly, which increased by 4.5% at 5 min, by another 5.5% at 30 min, and by an average of 1% from 5 to 30 min compared to the baseline values in the blocked side. The responses of the patients to the questions probed in the pinprick test and Bromage scale were fully compatible with the data obtained by the near-infrared spectroscopy (NIRS) method. *Conclusions:* StO2 monitoring may provide a useful instrument for rapid evaluation of the success of regional anesthesia in the upper extremity.

## 1. Introduction

Upper extremity peripheral blocks are a type of block commonly used in anesthesia practice. Over the last decade, the use of ultrasonography (USG) technology in peripheral nerve blocks has made it possible not only to determine the relationship between the peripheral anatomic structures and nerves but also to control the needle tip visually and to follow the spread of the injected local anesthetic (LA). USG guidance often increases the block success rates while decreasing the risk of complications [1,2]. Evidence of successful upper limb peripheral nerve block includes arterial vasodilatation, an increase in skin temperature, and an increase in blood flow along the brachial artery on the blocked side. In the evaluation of block success, clinicians generally use traditional evaluations that require oral confirmation from the patients such as the pinprick testing, warm/cold testing, and the Bromage scale [3]. However, it is ultimately at the discretion of the patient to decide whether the procedure should proceed with surgery independent of all these tests. On the other hand, in some patient groups such as elderly patients, disabled patients, and children, who are often difficult to communicate with in the operating room, the verbal tests abovementioned may not be appropriate for these groups. Therefore, there is need for an objective test that is fast, more reliable, and noninvasive for the effective evaluation of block success to be used as an alternative subjective measurement that is primarily based on the responses provided by the patients.

Near-infrared spectroscopy (NIRS) is a noninvasive device used to detect tissue oxygen saturation (StO2) based on microcirculation perfusion. NIRS detects the differences in oxygen saturation resulting from the changes in the tissue oxygen consumption or oxygen delivery [4]. Considering that peripheral nerve block leads to hemodynamic changes, we hypothesized that it might be possible to objectively evaluate the effectiveness of a block using NIRS. Accordingly, the first goal of this study was to detect changes of measuring StO2 with NIRS in the patients undergoing upper extremity surgery under infraclavicular block. The second goal of this study was to measure the efficacy and adequacy of the peripheral nerve block by measuring StO2 with NIRS.

## 2. Materials and Methods

### 2.1. Population

The study included 40 patients who underwent hand surgery under infraclavicular block at Harran University Research and Application Hospital Orthopedics and Cardiovascular Surgery departments. We calculated the sample size according to the results of the first 15 patients in the study. From these differences and assuming a two-tailed α value of 0.05 (sensitivity 95%) and a β value of 0.20 (study power: 80%, effect size: 0.80), we determined that at least 38 patients were required for our study by statistical software Package G Power (version 3.1.9.2; Franz Faul & Edgar Erdfelder, Trier, Germany). We decided to enroll 40 patients in this study. All the study procedures were performed after obtaining the necessary ethics committee approval and patient consents. Patients with contraindications for infraclavicular block, renal insufficiency, peripheral vascular disease, and an allergic reaction to LA or liver failure; patients on α- and β-blocker agent treatment; and pregnant and breastfeeding women were excluded from the study. Upper peripheral nerve block procedures were performed with ultrasonography (USG) by a single anesthesiologist and the measurements and tests were performed by another anesthesiologist.

### 2.2. Procedures

Patients received no sedative drugs on the morning of the surgery or in the operating room. Prior to the procedure, vascular access was established with an 18–20 G cannula through the back or antecubital area for each patient and fluid replacement was made with 0.9% isotonic NaCl solution (10 mL/kg). Electrocardiography (ECG), which is a standard monitoring method, was performed along with peripheral oxygen saturation (SpO2) and noninvasive arterial pressure (NIAP). Baseline systolic blood pressure (SBP), diastolic blood pressure (DBP), and heart rate (HR) were recorded. Noninvasive StO2 monitoring was performed using an InSpectra™ StO2 monitor (Hutchinson Technology Inc., Hutchinson, MN, USA). The probe was placed in the palm (thenar) of the upper extremity, the location where the blockade would be performed, and also in the palm (thenar) of the other extremity. Baseline StO2 and tissue hemoglobin index (THI) values were recorded. In the supine position, the head of the patient was turned to the nonblocked side of the area where the block was to be applied. The arm to which the block would be applied was abducted and placed on the chest in a flexion position. After the block area was covered in a sterilized manner following its disinfection with Poly(vinylpyrrolidone)-iodine complex, local anesthesia was induced with 1 cc 2% lidocaine. The linear probe was placed longitudinally to the area with the aid of a USG device (Esaote MyLab 30 Gold, linear probe, 10–18 MHz, Florence, Italy) in order to perform infraclavicular block. When the axial artery, vein, and brachial plexus cords became visible, 22 mL of LA solution (50% of a mixture of 2% lidocaine and 0.5% bupivacaine) was given using an 80 mm long 22G nerve stimulation needle by surrounding the artery with the posterior cord and the lateral cord in a crescent-like arrangement (Pajunk^®^, Geisingen, Germany).

### 2.3. Clinical Evaluation

After the application of the infraclavicular block, sensory and motor blocks were evaluated every 5 min for a total period of 30 min. In addition, SpO2, HR, NIAP, THI, and StO2 of both arms were recorded every 5 min. The total estimated time for the administration of the block was 30 min. At the 30th min, the sensory block level of the patient was evaluated using pinprick testing with a needle (0: no sensory block, 1: touch sensation, no pain, 2: no touch sensation and pain) and the motor block level was evaluated with the Bromage scale (0: ability of moving fingers and wrist, 1: poor motor and finger movement ability, 2: inability to move finger). In cases with no apparent change between the two assessments (i.e., pre- and postblock assessments), the block was accepted as unsuccessful. In cases with successful block, surgical procedures were continued as normal.

### 2.4. Statistical Analysis

Statistical analysis was performed using SPSS version 25.0 for Windows (Armonk, NY, USA: IBM Corp.) and PAST 3 (Øyvind Hammer, Natural History Museum, University of Oslo, Oslo, Norway) (Hammer, Ø., Harper, D.A.T., Ryan, P.D. 2001. Paleontological statistics). Normality of distribution was tested for one-variable data sets by Shapiro–Wilk test, whereas for multiple-variable data sets, normality of distribution was tested using Mardia’s multivariate skewness and kurtosis tests and the Doornik–Hansen omnibus test and the homogeneity of variance was tested using Box’s M test. The blocked and nonblocked extremities were compared in terms of StO2 and THI values using Mann–Whitney U test with Monte Carlo simulation. Repeated measurements of StO2, THI, HR, SpO2, SBP, and DBP were compared using Friedman’s Two-Way test with Monte Carlo simulation, followed by post hoc Dunn’s test. Quantitative variables were expressed as median (minimum–maximum) and standard deviation (SD) and categorical variables were expressed as percentages. A *p* value of <0.05 was considered significant.

### 2.5. Ethical Approval

The study was approved by Harran University Medical Faculty Ethics Committee (approval number: 2017.4.24) and was registered at the Australian New Zealand Clinical Trials Registry (ANZCTR) (ACTRN12617000688381).

## 3. Results

A total of 40 patients aged 18–64 years who underwent elective hand surgery under infraclavicular block were included in the study. Mean age was 37.33 ± 13.53 years. Demographic data of the patients are presented in Table 1. Block failure occurred in only 1 (2.5%) patient who was switched to general anesthesia. Since the number of patients with failed blocks was low, no statistical comparison could be performed between the patients with and without successful block.

Postanesthetic HR value decreased compared to its baseline value (*p* = 0.024), whereas all the postanesthetic HR values were similar (*p* = 0.761) (Figure 1). Pre- and postanesthetic SpO2 values were similar (*p* = 0.333). Postanesthetic SBP value decreased compared to its baseline value (*p* < 0.001), whereas all the postanesthetic SBP values were similar (*p* = 0.058). Postanesthetic DBP value decreased compared to its baseline value (*p* < 0.001), whereas all the postanesthetic DBP values were similar (*p* = 0.02) (Table 2).

Preanesthetic median StO2 values of the blocked side and nonblocked side were similar (*p* = 0.532), whereas the postanesthetic values of the blocked side were higher at the fifth minute and the following minute, measurements compared to the nonblocked side. Median StO2 value at baseline was 83% for the blocked side and 82% for the nonblocked side. No significant difference was found between the blocked side and the nonblocked side in terms of StO2 values. The median StO2 values increased significantly, which increased by 4.5% at 5 min, by another 5.5% at 30 min, and by an average of 1% from 5 to 30 min compared to the baseline values in the blocked side (Table 3, Figure 1). Similar to the variations in StO2, the THI median values of the blocked side and the nonblocked side were similar (*p* = 0.154), whereas the postanesthetic values of the blocked side were higher in all measurements compared to the nonblocked side (*p* < 0.05).

In the pinprick testing, 1 (2.5%) patient had no sensory block, 37 (92.5%) patients had touch sensation and no pain, and 2 (5%) patients had no touch sensation or pain. On the Bromage scale, however, one patient (2.5%) who did not undergo the block procedure was able to move fingers and wrist, whereas 37 (92.5%) patients had poor motor and finger movement ability and 2 (5%) patients were completely unable to move their fingers.

On the other hand, there was a patient who had neither sensory nor motor block and also had infraclavicular block failure, and in whom baseline StO2 value was 73.00% and the value at min 30 was 72.00%. Similarly, baseline THI value was 10.50 and the value at min 30 was 10.40. However, no significant increase was observed in both parameters. These findings indicate that although the patient had no complete sensory and motor block, the highest values of StO2 and THI were obtained in this patient.

## 4. Discussion

In this study, in the direction of our primary goal, the results indicated that the StO2 increase that occurred following a successful peripheral nerve block via the NIRS method can be considered as a noninvasive approach. Following the application of the block, in the direction of our secondary goal, the StO2 values increased in the blocked side starting from the first minutes. However, the block was unsuccessful in only one patient, in whom the StO2 values did not change and the Bromage scale and pinprick test were negative.

Various objective methods have been investigated in the literature which mainly focus on the blood flow following sympathetic blockade usually by relying on parameters and techniques such as the peripheral flow index, resistance index, perfusion index (PI), thermographic temperature measurement, skin temperature, finger photoplethysmography, noninvasive blood hemoglobin assessment, and the variability index [3,5,6,7,8,9,10]. However, none of these techniques are used as the golden standard for the assessment of block success and subjective methods remain the method of choice. Given the working principle of NIRS, which is an alternative to the abovementioned methods, this method can be the primary method of choice since it can be used for assessing arterial vasodilatation caused by sympathetic blockade after peripheral nerve block and also for measuring the increase in blood flow velocity and for evaluating block efficiency [11].

Literature reviews indicate that only a few studies have evaluated the efficacy of NIRS monitoring in the induction of peripheral nerve blocks [4,12,13]. Unlike these studies, our study is the first comprehensive study using NIRS in the administration of infraclavicular nerve block, which is a type of upper extremity peripheral nerve block. Moreover, unlike other studies, this is the first study on NIRS which compared block success and efficacy by using traditional methods and also measured the changes in StO2 on the blocked extremity. NIRS reflects the changes in vasomotor tone which varies among patients; therefore, block adequacy can only be assessed by determining the NIRS increase rate in comparison to the baseline value. Our study indicates that a significant increase in the StO2 value within the first 5 min may indicate an early block onset time (readiness for surgery), whereas an increase between 5 min and 30 min may indicate a delayed block onset time. When compared to other objective methods, NIRS appears to be a useful method since it achieved a successful early block onset time at min 5 in our study, which is the earliest block onset time in the literature. On the other hand, in finger photoplethysmography, which is also used as an objective method, the results of the analyses are very sensitive to patient movement and are also easily affected by various factors such as stress, anxiety, and age- and gender-related differences which induce peripheral vasoconstriction caused by sympathetic activation [14]. An important advantage of NIRS in the evaluation of peripheral nerve blocks is that the measured StO2 value is supported by the THI, which is a sign of vasodilatation and a hemoglobin signal strength metric used for determining whether the StO2 sensor is optimally positioned over the muscle. THI does not correlate with age, height, weight, body mass index, SBP, DBP, mean arterial pressure, or HR [15,16]. Another advantage of NIRS is that it is more practical, noninvasive, low-cost, and easily transportable with the patient, does not need consumables, is relatively easy to install and use with no operator dependence, can be used continuously throughout the perioperative period, leads to minimal movement-dependent artifacts, and does not require a separate probe for each patient.

As NIRS is an objective method, a sedative drug dose that will lead to conscious sedation can be used without affecting the usefulness and effectiveness of the technique [12]. In our study, no sedation was administered prior to the procedure in order to compare the NIRS with the pinprick test and Bromage scale. Moreover, the StO2 values of the blocked extremity were found to be compatible and in parallel with the pinprick test and Bromage scale. The responses of the patients to the questions probed in the pinprick test and Bromage scale were fully compatible with the data obtained by the NIRS method. In one patient with block failure, both the Bromage scale and pinprick test were negative and no change was found in the postanesthetic StO2 and THI values of the blocked extremity compared to the nonblocked extremity. During the block procedure, the patient was moving and highly agitated, which could be the primary reasons for block failure.

Our study was limited in several ways. First, our study was not a randomized and placebo-controlled study. Instead, the other extremity on which surgical procedure was not performed served as the control. Another limitation was the low number of failed blocks, for which no comparison could be performed. For this reason, we cannot give a satisfactory result on assessing the sensitivity, specificity, and predictive value of tissue oxygen saturations ipsilaterally to the block in our study. Finally, the volume of LA solution was remarkably high. An LA solution of 22 mL, which is commonly used as the standard dose in our hospital, was administered in the blocked area. In extremity peripheral nerve blocks, administering a low volume and a high concentration of LA is associated with a higher success rate and a shorter onset time than a high volume and a low concentration of solution. So, we consider that lower volumes and high concentrations of LA could have an impact on brachial plexus block and postoperative analgesia. Nevertheless, to our knowledge, there have been no studies reporting on LA dose, concentration of LA, and type of LA in the administration of NIRS.

## 5. Conclusions

The results indicated that NIRS is a useful predictor of infraclavicular block success. In addition, as a noninvasive and practical method, NIRS visualizes the changes that indicate a successful block even within the first 5 min. Therefore, in clinical practice, NIRS can provide a valuable tool for quickly assessing the success of upper extremity regional anesthesia. We also believe that the superiority of StO2 measurement with the NIRS method over the traditional methods can be substantiated in further large-scale studies that would involve specificity and sensitivity measurements.

## Figures and Tables

**Figure 1 medicina-55-00274-f001:**
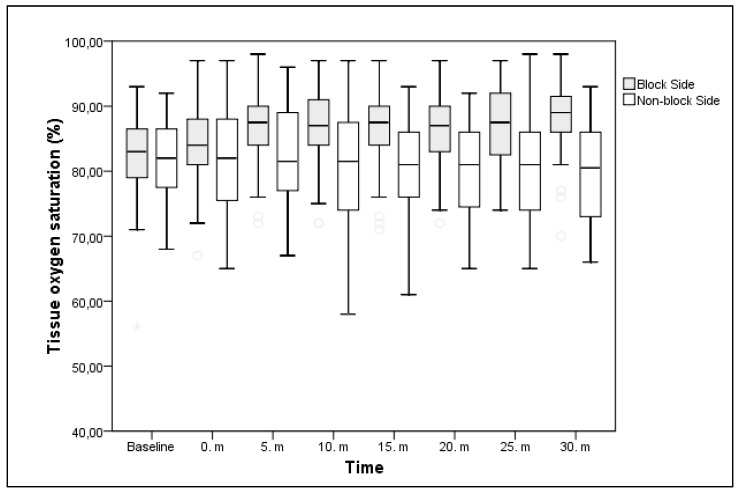
Changes in tissue oxygen saturation within 30 min in 5 min intervals after initiation of infraclavicular brachial plexus block in blocked side and nonblocked side. StO2 increased from the fifth minute in blocked side (*p* < 0.05) when block was successful. There was no significant change in nonblocked side compared to blocked side from the fifth minute.

**Table 1 medicina-55-00274-t001:** Demographic data.

	*n*	%
Gender		
Male	21	52.5%
Female	19	47.5%
**Comorbid disease**		
nonavailable	23	57.5%
available	17	42.5%
**Pinprick test**		
no sensory block	1	2.5%
touch sensation, no pain	37	92.5%
no touch sensation and pain	2	5.0%
**Bromage scale**		
ability of moving fingers and wrist	1	2.5%
poor motor and finger movement ability	37	92.5%
inability to move finger	2	5%

**Table 2 medicina-55-00274-t002:** Changes in vital findings over time.

Time (Minute)	HRMedian (Min/Max)	SpO2 Median (Min/Max)	SBP Median (Min/Max)	DBP Median (Min/Max)
Baseline	83 (53/113)	98.5 (94/100)	137.5 (106/201)	86.5 (48/118)
0	76 (53/104)	99 (94/100)	131 (84/203)	80 (41/113)
5	74 (49/103)	99 (94/100)	132.5 (84/194)	82 (46/113)
10	74 (46/113)	99 (92/100)	130.5 (89/201)	80.5 (47/109)
15	73 (55/98)	99 (93/100)	129.5 (81/192)	81.5 (50/113)
20	74 (55/100)	99 (90/100)	133 (92/196)	80.5 (56/119)
25	75.5 (53/99)	99.5 (94/100)	132.5 (94/193)	82 (55/108)
30	76.5 (53/100)	99 (95/100)	134.5 (102/198)	84 (58/112)
*p* **	0.024	0.333	<0.001	<0.001

HR: Heart rate (bpm), SBP = Systolic blood pressure (mmHg), DBP = Diastolic blood pressure (mmHg). ** Friedman Test (Monte Carlo), *p* < 0.05 was considered significant Post Hoc Test: Dunn’s Test.

**Table 3 medicina-55-00274-t003:** Changes in tissue oxygen saturation and total hemoglobin index in the blocked and nonblocked side.

	Blocked Side (%)	Nonblocked Side (%)	*p* *
Time (Minute)	Median (Min/Max)	Median (Min/Max)	
**Tissue Oxygen Saturation (%)**			
Baseline	83 (56/93)	82 (68/92)	0.532
0	84 (67/97)	82 (65/97)	0.012
5	87.5 (72/98)	81.5 (45/96)	0.006
10	87 (72/97)	81.5 (58/97)	<0.001
15	87.5 (71/97)	81 (59/93)	<0.001
20	87 (72/97)	81 (51/92)	<0.001
25	87.5 (74/97)	81 (54/98)	<0.001
30	89 (70/98)	80.5 (44/93)	<0.001
*p* **	<0.001	0.064	
**Tissue Hemoglobin Index**			
Baseline	10.7 (7.7/16.4)	10.5 (7.5/16.4)	0.154
0	11.3 (8.1/14)	11 (7.1/11.5)	0.004
5	11.7 (7.3/14.6)	11.3 (7.7/12.5)	0.012
10	11.2 (6.8/13.3)	10.2 (7.9/11.3)	0.001
15	12.4 (6.1/13.4)	10.3 (.6/13.1)	<0.001
20	13.8 (5/14.4)	11.5 (6.3/12.7)	<0.001
25	11.3 (5.2/17.1)	10.9 (6.8/12.8)	<0.001
30	10.9 (8.7/14.8)	10.4 (6.3/13.2)	0.013
*p* **	<0.001	<0.001	

* Mann–Whitney U test (Monte Carlo) ** Freidman test (Monte Carlo) Post Hoc Test: Dunn’s Test. Bl: Baseline.

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
