# Peer review of "Tissue Oxygen Saturation Change on Upper Extremities After Ultrasound-Guided Infraclavicular Brachial Plexus Blockade; Prospective Observational Study"

_medicina, 2019, doi:10.3390/medicina55060274_

Reviewer 1 Report

This is an interesting study that aimed to assess the effect of ultrasound-guided infraclavicular brachial plexus block on the tissue oxygen saturation. I have some comments and questions regarding the manuscript:

1.     The authors assessed tissue oxygen saturation on block and non-block side and provide information  on the values on Figure 1. The figure does not depict statistical significances at different time points. Please add them.

2.     Methods: What was the primary hypothesis of the study? Please descibe it explicitely. Why exactly 40 patients were included? Was there a power analysis performed? If so, what was the expectation based upon you performed the power analysis (if any). Please provide a detailed description on this.

3.     Among the limitation section it should be mentioned that the study was underpowered for assessing the sensitivity, specificity and the predictive value of tissue oxygen saturations ipsilaterally to the block. You will also need to include those patients in whom block was not complete.

Author Response

Dear reviewer,
Thank you for your letter and comments concerning our manuscript entitled “Tissue Oxygen Saturation Change on Upper Extremities After Ultrasound-Guided Infraclavicular Brachial Plexus Blockade: Prospective Observational Study”.  We have studied your comments carefully and made major correction which we hope meet with your approval. We answered your questions or comments in details in the following texts. 

Specific Comments:

Reviewer #1:

Comment 1:  The authors assessed tissue oxygen saturation on block and non-block side and provide information  on the values on Figure 1. The figure does not depict statistical significances at different time points. Please add them

Respond 1:  Thank you for possitive suggestion.  Changes in tissue oxygen saturation within 30 minutes in 5 min intervals after initiation of infraclavicular brachial plexus block in block side and non -blocked side.  StO2 increased from the 5th minutes in block side (p<0,05) when block was successful. There was no significant change in non-block side compared block side from the 5th minutes.In the figure, the above-mentioned required text is added.

Comment 2:  Methods: What was the primary hypothesis of the study? Please descibe it explicitely. Why exactly 40 patients were included? Was there a power analysis performed? If so, what was the expectation based upon you performed the power analysis (if any). Please provide a detailed description on this.

Respond 2: Thank you for the comments.  We calculated the sample size according to the results of the first fifteen patients in the study. From these differences and assuming a two-tailed α value of 0.05 (sensitivity 95%) and a β value of 0.20 (study power: 80%, effect size: 0.80), we determined that at least 38 patients were required for our study by statistical software Package G Power (version 3.1.9.2 ; Franz Faul & Edgar Erdfelder, Trier, Germany). We decided to enrol 40 patients in this study.

Comment 3  Among the limitation section it should be mentioned that the study was underpowered for assessing the sensitivity, specificity and the predictive value of tissue oxygen saturations ipsilaterally to the block. You will also need to include those patients in whom block was not complete.

Respond 3:   Thank you for possitive suggestion.  limitation section was revised according to the recommendations. There are no patients in whom we interrupt the block process. however, there was only one patient who underwent general anesthesia because of a failed block. it is also mentioned in the method section.

We really hope theses modification can meet with your approval. Thank you very much.

Reviewer 2 Report

Authors suggested that the tissue oxygen saturation monitoring produces faster, reliable and noninvasive for the effective onset of  blockade in the upper extremity.

The problem is well described, the hypotheses are well-stated, and the analysis is appropriate.

I have no hesitation in recommending the paper for publication. I think that the following would help readers:

How the differences in the dose, volume, and type of local anesthetic used, the primary outcome variable and criteria used to define the onset time,  may have played a role compared to other studies. Please comment on that in the manuscript. 

Currently, there is no standardized definition for “onset time” ( the onset of sensory blockade, or the onset of sensory-motor blockade or “readiness for surgery”).  Please comment on that in the manuscript. 

Ethical approval is repeated twice in text 

Author Response

Dear reviewer,
Thank you for your letter and comments concerning our manuscript entitled “Tissue Oxygen Saturation Change on Upper Extremities After Ultrasound-Guided Infraclavicular Brachial Plexus Blockade: Prospective Observational Study”.  We have studied your comments carefully and made major correction which we hope meet with your approval. We answered your questions or comments in details in the following texts. 

Specific Comments:

Reviewer #2:

Comment 1:  How the differences in the dose, volume, and type of local anesthetic used, the primary outcome variable and criteria used to define the onset time,  may have played a role compared to other studies. Please comment on that in the manuscript.

Respond 1:  In extremity peripheral nerve blocks, administering a low volume and a high concentration of local anesthetic is associated with a higher success rate and a shorter onset time than a high volume and a low concentration of solution. Nevertheless, to our knowledge, there have been no studies reporting on LA dose, concentration of LA and type of LA in the administration of NIRS. We also mentioned this in the limitation section of our manuscript.

Comment 2:  Currently, there is no standardized definition for “onset time” ( the onset of sensory blockade, or the onset of sensory-motor blockade or “readiness for surgery”).  Please comment on that in the manuscript. 

Respond 2: Thank you for possitive suggestion.  We used the definition of onset time to be readiness for surgery. We have also stated this in the article.

Comment 3  Ethical approval is repeated twice in text 

Respond 3:   Thank you for possitive suggestion.  We made necessary corrections on ethical approval.

We really hope theses modification can meet with your approval. Thank you very much.